# Bayesian Inference of State-Level COVID-19 Basic Reproduction Numbers across the United States

**DOI:** 10.3390/v14010157

**Published:** 2022-01-15

**Authors:** Abhishek Mallela, Jacob Neumann, Ely F. Miller, Ye Chen, Richard G. Posner, Yen Ting Lin, William S. Hlavacek

**Affiliations:** 1Department of Mathematics, University of California, Davis, CA 95616, USA; amallela@ucdavis.edu; 2Department of Biological Sciences, Northern Arizona University, Flagstaff, AZ 86011, USA; jn382@cornell.edu (J.N.); efm46@nau.edu (E.F.M.); richard.posner@nau.edu (R.G.P.); 3Department of Mathematics and Statistics, Northern Arizona University, Flagstaff, AZ 86011, USA; ye.chen@nau.edu; 4Los Alamos National Laboratory, Information Sciences Group, Computer, Computational and Statistical Sciences Division, Los Alamos, NM 87545, USA; yentingl@lanl.gov; 5Los Alamos National Laboratory, Theoretical Biology and Biophysics Group, Theoretical Division, Los Alamos, NM 87545, USA

**Keywords:** mathematical model, coronavirus disease 2019 (COVID-19), basic reproduction number, herd immunity, Bayesian inference

## Abstract

Although many persons in the United States have acquired immunity to COVID-19, either through vaccination or infection with SARS-CoV-2, COVID-19 will pose an ongoing threat to non-immune persons so long as disease transmission continues. We can estimate when sustained disease transmission will end in a population by calculating the population-specific basic reproduction number ${ℛ}_{0}$, the expected number of secondary cases generated by an infected person in the absence of any interventions. The value of ${ℛ}_{0}$ relates to a herd immunity threshold (HIT), which is given by $1-1/{ℛ}_{0}$. When the immune fraction of a population exceeds this threshold, sustained disease transmission becomes exponentially unlikely (barring mutations allowing SARS-CoV-2 to escape immunity). Here, we report state-level ${ℛ}_{0}$ estimates obtained using Bayesian inference. Maximum a posteriori estimates range from 7.1 for New Jersey to 2.3 for Wyoming, indicating that disease transmission varies considerably across states and that reaching herd immunity will be more difficult in some states than others. ${ℛ}_{0}$ estimates were obtained from compartmental models via the next-generation matrix approach after each model was parameterized using regional daily confirmed case reports of COVID-19 from 21 January 2020 to 21 June 2020. Our ${ℛ}_{0}$ estimates characterize the infectiousness of ancestral strains, but they can be used to determine HITs for a distinct, currently dominant circulating strain, such as SARS-CoV-2 variant Delta (lineage B.1.617.2), if the relative infectiousness of the strain can be ascertained. On the basis of Delta-adjusted HITs, vaccination data, and seroprevalence survey data, we found that no state had achieved herd immunity as of 20 September 2021.

## 1. Introduction

Vaccines to protect against coronavirus disease 2019 (COVID-19) became available in the United States (US) in December 2020 [1]. In the US, as of 20 September 2021, 181,728,072 persons have been fully vaccinated (55% of the total population), an additional 30,307,256 persons have been partially vaccinated, and an uncertain number of persons have acquired immunity through infection [2]. The entire US population does not need to be vaccinated to end sustained COVID-19 transmission because of the phenomenon of herd immunity [3], which is reached when a critical fraction of the population becomes immune. This fraction is called the herd immunity threshold (HIT).

The HIT for a population relates to the basic reproduction number, ${ℛ}_{0}$, as follows [3]: HIT $=1-1/{ℛ}_{0}$. ${ℛ}_{0}$ is defined as the expected number of secondary infections arising from a primary case in the absence of any immunity or intervention. As is well known, ${ℛ}_{0}$ and HIT are population-specific [4,5], which means that the effort required to control the local COVID-19 epidemic may vary from community to community. However, knowledge of the HIT for a given region is insufficient to determine when disease transmission within the region will end. One also needs to know the fraction of the population that has immunity. Estimating the immune fraction is difficult, because we cannot simply count the number of persons who have been vaccinated or the number of persons detected to be infected. Immunity is acquired not only through vaccination but also through infection [6], and case detection is imperfect. Insight into the immune fraction can be obtained from seroprevalence surveys, which use blood tests to identify persons who have antibodies against the SARS-CoV-2 virus (acquired through vaccination or infection).

Various estimates of ${ℛ}_{0}$ for transmission of COVID-19 have been provided in the literature [7]. The estimates that have received the most attention are those given for China and Italy [8,9,10,11,12], which were among the first regions to be impacted by COVID-19. However, the relevance of these estimates for populations within the US (or elsewhere outside of China and Italy) is unclear. Several studies have estimated ${ℛ}_{0}$ for the US at the national level [13,14,15], the state level [16,17,18], and the county level [19,20]. The usefulness of a national estimate is unclear given the heterogeneity of the US, and none of the county-level estimates are comprehensive. Some state-level estimates are also incomplete [16,18]. Because responses to COVID-19 within the US have been and continue to be driven mainly by governors of US states [21], we undertook a study to generate comprehensive state-level ${ℛ}_{0}$ estimates through Bayesian inference. With this approach, we were able to quantify uncertainty in each estimate through a parameter posterior distribution.

In earlier work, we developed a compartmental model for COVID-19 transmission dynamics that reproduces surveillance data and generates accurate forecasts for the 15 most populous metropolitan statistical areas (MSAs) in the US [22]. Here, for each of the 50 states, we found a state-specific parameter posterior conditioned on this model from state-level COVID-19 surveillance data available from 21 January to 21 June 2020 [23]. From these parameter posteriors, we then obtained region-specific ${ℛ}_{0}$ and HIT posteriors and maximum a posteriori (MAP) estimates. The MAP estimates for HITs together with other data—vaccination tracking data [24], serological survey data [25,26], and quantitative estimates of the increased transmissibility of the recently introduced SARS-CoV-2 variant Delta (lineage B.1.617.2) [27,28]—provide insight into the progress of each state toward herd immunity.

## 2. Materials and Methods

### 2.1. Model

To obtain regional ${ℛ}_{0}$ and HIT estimates, we used a compartmental model developed previously for the purpose of forecasting COVID-19 disease incidence [22]. This model, which is capable of making accurate forecasts [22], is a COVID-19-specific elaboration of the classic SEIR model [29] that accounts for effects of nonpharmaceutical interventions, including social distancing. Consideration of nonpharmaceutical interventions is important because the widespread adoption of such interventions began in the US around 13 March 2020 [30], a time roughly coincident with the start of sustained community transmission of COVID-19 in many parts of the US (see below). We found region-specific parameterizations that allow the model to reproduce surveillance data (daily reports of new confirmed COVID-19 cases) available for each region of interest over a defined period (e.g., 21 January to 21 June 2020). The model is able to account for a variable number of social-distancing periods. We considered versions of the model accounting for one, two, and three social-distancing periods. The number of social-distancing periods deemed best (i.e., to provide the most parsimonious explanation of the data) for a given time period was determined using the model selection procedure described by Lin et al. [22]. As in the study of Lin et al. [22], the model has 14 parameters with universal fixed values (applicable to all regions). The model also has 3(n+1)+3 parameters with region-specific adjustable values determined through Bayesian inference, where n+1 denotes the number of social-distancing periods. In this study, for a given region, we censored case-reporting data whenever the cumulative reported case count was less than 10 cases. We also specified the onset time of the first social-distancing period, σ, as the earliest day on which the cumulative reported case count was 200 cases or more. A full description of model parameters is given in Lin et al. [22].

### 2.2. Simulations

Each region-specific model consists of a coupled system of ordinary differential equations (ODEs), which are given by Lin et al. [22]. The ODEs were numerically integrated using the SciPy [31] interface to LSODA [32] and the BioNetGen [33] interface to CVODE [34]. Python code was converted to machine code using Numba [35]. The initial conditions were determined as in Lin et al. [22].

### 2.3. Calculation of Epidemic Parameters ${ℛ}_{0}$ and λ

To find the basic reproduction number ${ℛ}_{0}$, we considered a reduced form of the model of Lin et al. [22], which is given in Equations (1)–(8) of the Appendix A. The reduced model omits consideration of interventions, including social distancing, quarantine, and self-isolation, which are all considered in the full model. From the reduced model, we derived an expression for ${ℛ}_{0}$ by applying the next-generation matrix method [36]. In this procedure, ${ℛ}_{0}$ is determined as the spectral radius of the so-called next-generation matrix. Denoting this matrix as N, the (i,j) entry of N is the expected number of new infections in the ith compartment produced by persons initially in the jth compartment. The expression for ${ℛ}_{0}$ given in the Results section below was obtained using Mathematica [37]. The matrix N was obtained using Mathematica’s LinearSolve function, and ${ℛ}_{0}$ was computed as the dominant eigenvalue of N.

To characterize the initial rate of exponential growth for a local epidemic within a given region, we computed the epidemic growth rate λ as the dominant eigenvalue of the Jacobian of the reduced model linearized at the disease-free equilibrium [38]. The derivation of λ is provided in the Appendix A.

### 2.4. Bayesian Inference

To infer region-specific values of adjustable model parameters (and ${ℛ}_{0}$ and HIT estimates), we followed the Bayesian inference approach of Lin et al. [22]. In inferences, we used all region-relevant confirmed COVID-19 case-count data available in the GitHub repository maintained by *The New York Times* newspaper [23] for the period starting on 21 January 2020 and ending on 21 May 2020, 21 June 2020, or 21 July 2020 (inclusive dates). The first case in the US was reported on 21 January 2020 [39]. We focused on early surveillance data (vs. all available surveillance data) so as to characterize COVID-19 transmission within populations that are nearly wholly susceptible. Markov Chain Monte Carlo (MCMC) sampling was performed using the Python code of Lin et al. [22] and a new release of PyBioNetFit [40], version 1.1.9, which includes an implementation of the adaptive MCMC method used in the study of Lin et al. [22]. Inference job setup files for PyBioNetFit, including data files, are provided for each of the 50 states online (https://github.com/lanl/PyBNF/tree/master/examples/Mallela2021States (accessed on 19 September 2021). Results from both methods were found to be consistent (Appendix A). To ensure that the MCMC sampling procedures converged, we visually inspected trace plots for log-likelihood (Appendix A), parameters (Appendix A) and pairs plots (Appendix A). We also performed simulations using maximum likelihood estimates (MLEs) for parameter values to assess the agreement of the simulations with the training data (Appendix A).

The maximum a posteriori (MAP) estimate of a parameter is the value of the parameter corresponding to the peak of its marginal posterior distribution, where probability density is highest. Because we assumed a proper uniform prior distribution for each of the adjustable parameters, as in the study of Lin et al. [22], the MAP estimates are MLEs.

## 3. Results

### 3.1. Bayesian Uncertainty Quantification

Following the Bayesian inference approach of Lin et al. [22], we quantified uncertainty in the predicted trajectories of confirmed case counts for all 50 states, using data from 21 January to 21 June 2020. As illustrated in Figure 1 for the states of New Jersey, Wyoming, Florida, and Alaska, we find that each region-specific model parameterized on the basis of our MCMC sampling procedure reproduces the corresponding surveillance data over the period of interest. Results for the remaining states are shown in Appendix A. At the end of each MCMC sampling procedure, we obtained a marginal posterior distribution for β (the rate constant in the model for disease transmission) which provides a probabilistic characterization of region-specific SARS-CoV-2 transmissibility. If the marginal posterior was narrow, we have high confidence in the MAP estimate of β; if it is wide, we had less confidence in its value. Each state-specific marginal posterior yielded a MAP estimate for β. 

We can propagate the uncertainty in β into uncertainty in ${ℛ}_{0}$ and HIT estimates, using the formula for ${ℛ}_{0}$ given below and $\mathrm{HIT}=1-1/{ℛ}_{0}$. In Figure 2, we show marginal posterior distributions for ${ℛ}_{0}$ and HIT for the states of New Jersey, Wyoming, Florida, and Alaska. We provide MAP estimates of the model parameters for all states in Appendix A. The model parameters were found to be identifiable in practice (we had no proof of identifiability). MAP estimates for ${ℛ}_{0}$ and HIT for all 50 states are provided in Appendix A. These tables also provide 95% credible intervals. These estimates characterize the infectiousness of SARS-CoV-2 ancestral strains in each region of interest.

### 3.2. Region-Specific Basic Reproduction Numbers and Herd Immunity Thresholds

To calculate the herd immunity threshold (HIT) for a specific region, we need to know the corresponding region-specific value of the basic reproduction number ${ℛ}_{0}$, which is given by the following formula (obtained as described in Materials and Methods and Appendix A):(1)$${ℛ}_{0}=\beta \times \left(\frac{1-f_{A}}{c_{I}}+\frac{f_{A}\rho_{A}}{c_{A}}+\frac{\left(m-1\right)\rho_{E}}{k_{L}}\right)$$
where β characterizes the rate of transmission attributable to contacts between persons who are not protected by social distancing, $f_{A}$ denotes the fraction of infected persons who never develop symptoms (i.e., the fraction of asymptomatic cases), $c_{A}$ characterizes the rate at which asymptomatic persons recover during the immune clearance phase of infection, $c_{I}$ characterizes the rate at which symptomatic persons with mild disease recover or progress to severe disease, $\rho_{E}$ is a constant characterizing the relative infectiousness of presymptomatic persons compared to symptomatic persons (with the same behaviors), $\rho_{A}$ is a constant characterizing the relative infectiousness of asymptomatic persons compared to symptomatic persons (with the same behaviors), *m* denotes the number of stages in the incubation period, and $k_{L}$ characterizes disease progression from one stage of the incubation period to the next and ultimately to an immune clearance phase. The value of ${ℛ}_{0}$ depends on one inferred region-specific parameter, β, and seven fixed parameters, which have values taken to be applicable for all regions (i.e., $f_{A},c_{A},c_{I},\rho_{E},\rho_{A},k_{L}$, and m). Estimates of these fixed parameters were taken from Lin et al. [22].

The SARS-CoV-2 variant Delta (lineage B.1.617.2) has been estimated to be 1.64 times more infectious than variant Alpha (lineage B.1.1.7) [28], which has been estimated to be 1.50 times more infectious than ancestral strains [27]. Assuming that Delta is the dominant circulating SARS-CoV-2 strain throughout the US (as of 20 September 2021) and that β for Delta is 1.64×1.50=2.46 times greater than β for ancestral strains (with other parameters in Equation (1) remaining the same), the MAP estimate of the Delta-adjusted ${ℛ}_{0}$ ranges from 5.6 for Wyoming to 18 for New Jersey (from the multiplier given above and Appendix A). The population-weighted Delta-adjusted ${ℛ}_{0}$ for the US is 12. These estimates indicate that the herd immunity threshold (HIT) for the Delta variant of SARS-CoV-2 ranges from 82% to 94%.

### 3.3. Estimates of Initial Region-Specific Epidemic Growth Rates

HIT estimates were directly determined by estimates of the basic reproduction number, which were related to the initial growth rate of the epidemic in a given region. Here, our ${ℛ}_{0}$ estimates were conditioned on a compartmental model that has been parameterized to reproduce case-reporting data available for each region over a five-month period (21 January to 21 June 2020). We can use parameter estimates obtained for each region to calculate the initial epidemic growth rate λ, which is directly comparable to early surveillance data (Figure 3 and Appendix A). We provide MAP estimates and 95% credible intervals for λ, ${ℛ}_{0}$, and HIT for selected states in Table 1. MAP estimates and 95% credible intervals for λ, ${ℛ}_{0}$, and HIT for all states are provided in Appendix A. These estimates are based on the state-specific marginal posteriors for the parameter β of our compartmental model. State-specific MAP estimates and 95% credible intervals for β (and other adjustable model parameters) are given in Appendix A. As can be seen (e.g., in Figure 3), our λ estimates are consistent with early case reporting data during the exponential takeoff phase of disease transmission.

### 3.4. Sensitivity of β to the Surveillance Data Used in Inference

For each state, we used surveillance data available from 21 January to 21 June 2020 to infer the MAP estimate of β (and the values of the other region-specific adjustable model parameters). This time window encompasses the onset time σ for all 50 states (Figure 4), which ranged from 10 March to 7 April 2020. Recall that σ is a region-specific parameter of the model of Lin et al. [22], which we take as the first time at which the cumulative confirmed case count for a given state was 200 or more. The value of σ provides a rough estimate of the start of sustained community transmission. To check the robustness of the MAP estimates for β to variations in training data, we performed a sensitivity analysis wherein we inferred β using data collected over three distinct periods in 2020, namely: (1) 21 January to 21 May, (2) 21 January to 21 June, and (3) 21 January to 21 July 2020. By visualizing our estimates with a rank order plot (Figure 5) and conducting pairwise two-sample Kolmogorov–Smirnov tests [41], we found that the 4-, 5-, and 6-month training datasets yielded estimates for β that were not statistically significantly different from each other. The MAP estimates for β obtained using the 4-, 5-, and 6-month datasets are listed in Appendix A. We assessed sensitivity by computing the relative error between the β estimates obtained from the 5-month dataset and the average β estimate over all datasets considered. We found that none of the state-level MAP estimates for β showed sensitivity (i.e., a relative error exceeding 100% in magnitude) to variations in the training data (Appendix A). The largest relative error was 12% (for Kansas).

### 3.5. Global Asymptotic Stability of the Disease-Free Equilibrium

The model of Lin et al. [22] has a globally asymptotically stable disease-free equilibrium (DFE) if ${ℛ}_{0}<1$, which can be deduced by following the approach of Shuai and van den Driessche [42]. As a consequence, the model predicts that the epidemic will be extinguished as the system dynamics are attracted to the DFE.

To confirm that the model behaves as expected around the HIT, we conducted a perturbation analysis for the states of New York (Figure 6A,B) and Washington (Figure 6C,D). We simulated disease dynamics starting from an arbitrarily chosen initial condition near the HIT number of persons, $S_{h}$, given by the following formula: $S_{h}=\mathrm{HIT}\times S_{0}$, where $S_{0}$ denotes the population size of the region considered. We defined the size of our perturbation as $\varepsilon =0.2\times S_{h}$ for Figure 6A,C and as $\varepsilon =-0.2\times S_{h}$ for Figure 6B,D. The initial condition was $S_{0}-S_{h}-1+\varepsilon$ susceptible persons, 1 infected person, and $S_{h}-\varepsilon$ recovered persons. As expected, for $S_{h}<\mathrm{HIT}\times S_{0}$ (Figure 6A,C), the number of infectious persons grows over time, whereas for $S_{h}>\mathrm{HIT}\times S_{0}$ (Figure 6B,D), the number of infectious persons decays over time.

In the two scenarios considered above (i.e., introduction of an infected person into a disease-free population with or without herd immunity), the rate at which disease burden changes is sensitive to different factors (Figure 7). As illustrated in Figure 7A, the rate at which disease burden decreases in a population with herd immunity (as in the scenario considered in Figure 6B,D) depends sensitively on the duration of the incubation period. As illustrated in Figure 7B, the rate at which disease burden increases (as in the scenario considered in Figure 6A,C) depends sensitively on the size of the subpopulation of susceptible persons.

### 3.6. Progress toward Herd Immunity

From our state-specific HIT estimates and other information (discussed below), we were able to calculate percent progress toward herd immunity for each state (Figure 8, Appendix A). We estimated the percent progress of each state’s population toward herd immunity, 𝒫∈[0%,100%], using the following equation (the derivation of which is given in the Appendix A):
(2)$$𝒫\equiv \left(\varepsilon_{v}\left(1-f_{r}\right)f_{v}+\varepsilon_{r}f_{r}\right){\left(1-\frac{1}{Y_{Delta}{ℛ}_{0}}\right)}^{-1}\times 100\%$$
where ${ℛ}_{0}$ is the population-specific basic reproduction number that we estimated for ancestral strains (Appendix A), $Y_{Delta}$ is a multiplier that accounts for the increased transmissibility of SARS-CoV-2 variant Delta, $f_{r}$ denotes the fraction of the population with immunity acquired through infection, $f_{v}$ is the fraction of the population that has been vaccinated [24], $\varepsilon_{r}$ is the fraction of infected persons who are protected against productive infection (i.e., an infection that can be transmitted to others), and $\varepsilon_{v}$ is the fraction of vaccinated persons who are protected against productive infection. Recall that we use $Y_{Delta}=2.46$ [27,28]. We estimate that $\varepsilon_{r}=1.0$ [43] and $\varepsilon_{v}=0.66$ [44]. We obtain four different estimates for $f_{r}$ as follows. In the first case, we obtain $f_{r}$ as the cumulative number of detected cases within a population divided by the population size. In the second case, we adjust our previous estimate for $f_{r}$ by a multiplier of 5.8 [45]. In other words, we assume that the true disease burden is 5.8 times higher than the detected number of cases. In the third case, we obtain $f_{r}$ as the fraction of the population that has been infected according to the latest serological survey results reported online at Ref. [25]. In the fourth case, we assume $f_{r}=f_{r,0}/\left(1-f_{A}\right)$, where $f_{r,0}$ denotes the estimate of seroprevalence in a given region and $f_{A}$ denotes the fraction of all cases that are asymptomatic. With this approach, we are assuming that asymptomatic cases are not detected in serological testing [46]. We adopt the estimate of Lin et al. [22] that $f_{A}=0.44$. 

As can be seen in Figure 8C, which is based on case reporting data, 18 of the 50 states have reached herd immunity. However, in Figure 8D, which is based on serological survey data, none of the states have reached herd immunity. South Dakota is closest to herd immunity, with the 84% of immune persons required for herd immunity. Idaho is furthest from herd immunity, with 45% of the immune persons required for herd immunity. The mean (median) progress toward herd immunity, across all states, is 63% (63%).

In Figure 9, we show the fraction of each state’s population that has been vaccinated and the fraction that is eligible for vaccination based on data available as of 20 September 2021. Vaccination data was taken from Ref. [24]. We assumed that only persons 18 years or older were eligible for vaccination. Age data were taken from Ref. [47]. In Figure 9, we also show Delta-adjusted HITs from Appendix A. As can be seen, vaccine coverage is below that required for herd immunity (in the face of Delta) in all cases, even if we take vaccines to provide sterilizing immunity in 100% of cases. For example, vaccine coverage for New Jersey (Wyoming) is 63% (41%) (Figure 9) and the corresponding Delta-adjusted HIT is 94% (82%) (Figure 9, Table 1, Appendix A). It seems that herd immunity cannot be reached through vaccination alone.

## 4. Discussion

One of our most important findings is a quantification of how COVID-19 transmissibility, in terms of the basic reproduction number ${ℛ}_{0}$, varies across the 50 US states. The MAP value of ${ℛ}_{0}$ for ancestral strains of SARS-CoV-2 ranges from 2.3 for Wyoming to 7.1 for New Jersey. The population-weighted mean for the US is 4.7. These estimates indicate that the herd immunity threshold (HIT) for the Delta variant of SARS-CoV-2 ranges from 82% to 94%, assuming that Delta is 2.46 times more transmissible than ancestral strains. The uncertainty in each ${ℛ}_{0}$ estimate was quantified: 95% credible intervals are indicated in Figure 5. The 95% credible intervals for ancestral HIT estimates are given in Appendix A. Because we can estimate the relative effort required to reach herd immunity across the US (in terms of HIT), resources for vaccination campaigns can be targeted to those areas where it is more difficult to achieve herd immunity.

Our ${ℛ}_{0}$ and HIT estimates differ from estimates given in previous studies. For example, various researchers derived point estimates for ${ℛ}_{0}$ from data using tools from time-series analysis, without assuming an underlying mechanistic model [13,15]. These tools depend on slope estimation and thus can be expected to depend sensitively on noise and errors in early case-reporting data. Ives and Bozzuto [16] provided state-level estimates for ${ℛ}_{0}$ (in 36 states), and Fellows et al. [17] used a Bayesian framework to obtain state-level estimates for ${ℛ}_{0}$ (in all 50 states). For the 30 states that are considered in Ives and Bozzuto [16], Fellows et al. [17], Milicevic et al. [18], and the present study, our estimates for ${ℛ}_{0}$ were most similar to those of Milicevic et al. [18] (Appendix A). Milicevic et al. [18] provided state-level ${ℛ}_{0}$ point estimates (for 45 states) that are statistically consistent with our MAP estimates of ${ℛ}_{0}$ for ancestral strains of SARS-CoV-2. The main points of difference between these earlier studies and the present study are as follows. Our ${ℛ}_{0}$ and HIT estimates were obtained from a model consistent with new case-reporting data, as illustrated in Figs 1 and 3. We were able to provide estimates for all 50 states (Figure 5, Appendix A), and we were able to obtain a Bayesian quantification of the uncertainty in each estimate (Figure 5, Appendix A).

In the face of Delta, the estimates of Figure 8C (based on case reporting data) suggest that a majority of states have yet to achieve herd immunity, and the estimates of Figure 8D (based on serological survey results) suggest that no state in the US has achieved herd immunity as of 20 September 2021. In either case, persons in the US lacking immunity are still at risk [48]. The perspective provided by Figure 8D is consistent with the study of Moghadas et al. [49] indicating that only 62% of persons in the US had some form of immunity as of 15 July 2021 (either through infection or vaccination). Given that the percentage of immune persons required for herd immunity according to Figure 8D ranges from 84% for South Dakota to 45% for Idaho (Figure 8D) ~20 months (counting from January 2020) into the COVID-19 pandemic and ~9 months after vaccines became widely available, it seems that this situation will persist for months, if not years.

How can the US accelerate the approach to herd immunity (if herd immunity is even possible)? Policies that encourage infection of children and vaccinated persons who have healthy immune systems may be rationalized because such persons seem to be well-protected against severe (but not mild) disease [50] and infected persons seem to have greater protection against productive infection [43]. However, this approach has obvious drawbacks, starting with the risks of infection. Another is that non-immune persons may not be able to self-identify as such. Unfortunately, it seems that we cannot rely on currently available vaccines to stop community transmission. Delta-adjusted HITs are mathematically impossible to achieve through vaccination alone because these HITs are close to 1 (Appendix A) and vaccine protection against productive infection is imperfect (i.e., $\varepsilon_{v}$ is significantly less than 1) [44]. This situation is exacerbated by the emergence of the SARS-CoV-2 variant Omicron (lineage B.1.1.529) [51], which has been estimated to be roughly 2 to 4 times more transmissible than Delta [52,53,54]. Other factors influencing the feasibility of herd immunity are waning immunity [55,56,57], limited vaccine uptake, and vaccine eligibility (Figure 9). Thus, use of variant-targeted vaccines may be needed to achieve herd immunity and to minimize COVID-19 impacts.

As is well-known, population features, not just pathogen features, affect the value of ${ℛ}_{0}$ [58]. These features potentially include numerous biological, sociobehavioral, and environmental factors, such as age, physical fitness, social network structure, population density, and aspects of the built environment. Variations in these features across regions can give rise to spatial heterogeneity in β and ${ℛ}_{0}$, although not in immediately obvious ways. One benefit of our comprehensive state-level ${ℛ}_{0}$ estimates is that they quantify how differences in population features across the US influence the spread of an aerosol-transmitted virus [59,60]. This information, by identifying the regions in the US where transmission is likely to be highest, could be useful in preparing for and responding to future pandemics caused by viruses similar to SARS-CoV-2. Disease transmission can be reduced through nonpharmaceutical interventions, such as early detection and isolation of infected persons [61,62,63].

Our study has several notable limitations. Our HIT estimates are potentially biased downward because of general awareness within the US of the impacts of COVID-19 in other countries (e.g., China and Italy), which could have resulted in a fraction of the US population changing their behaviors to protect themselves from COVID-19 before the start of the local epidemic. In addition, our estimation of percent progress toward herd immunity crucially depends on the seroprevalence estimates of the true disease burden. These estimates are associated with some uncertainty [64,65,66]. As illustrated in Figure 8, percent progress toward herd immunity is underestimated if serological tests fail to detect all cases of infection. The reader must also be cautioned that our analysis depends on a number of assumptions. For example, we considered a compartmental model in which populations are taken to be well-mixed and to lack age structure. This is clearly a simplification. More refined estimates could be obtained by making the model more realistic, but this would have the drawback of increasing the complexity of inference, which at some point would make inference impracticable.

## Figures and Tables

**Figure 1 viruses-14-00157-f001:**
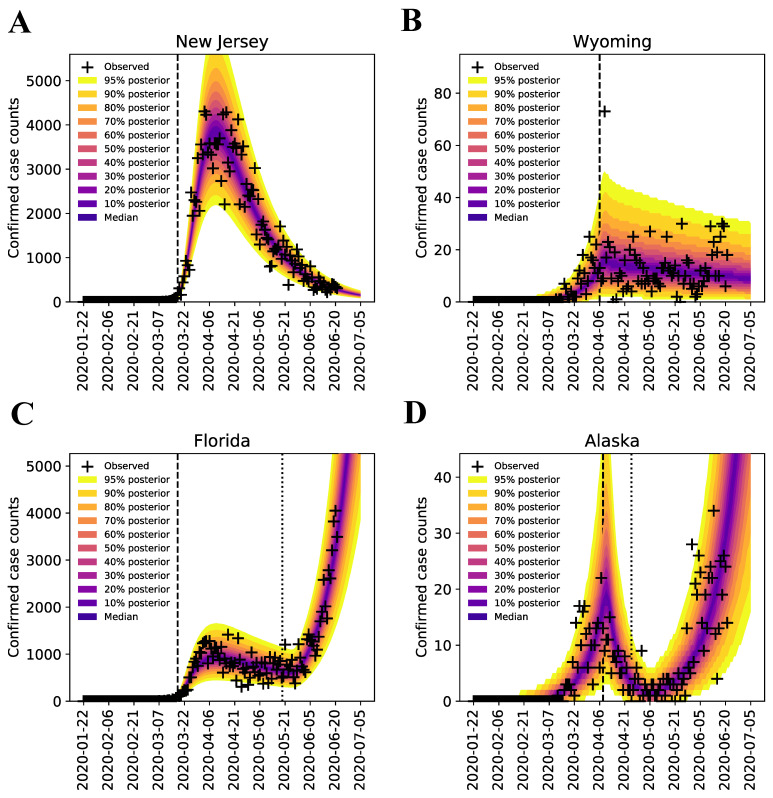
Bayesian predictive inferences for daily confirmed case counts of COVID-19 in (**A**) New Jersey (**B**) Wyoming (**C**) Florida (**D**) Alaska, from 21 January to 21 June 2020 (inclusive dates). The compartmental model [22] accounts for an initial social distancing period followed by n additional periods. We considered n=0, 1 and 2 and selected the best n using the model selection procedure of Lin et al. [22]. Plus signs indicate daily case reports. The shaded region indicates the prediction uncertainty and inferred noise in the detection of new cases. The color-coded bands within the shaded region indicate the median and different credible intervals (e.g., the dark purple band corresponds to the median, the band with lightest shade of yellow corresponds to the 95% credible interval, and gradations of color between these two extremes correspond to different credible intervals, as indicated in the legend). In each panel, the vertical broken line indicates the onset time of the first social-distancing period. For states with n=1 (Alaska and Florida), there is an additional broken line, which indicates the onset time of the second social-distancing period. The model was used to make forecasts of new case detection for 14 days after 21 June 2020. The last prediction date was 5 July 2020.

**Figure 2 viruses-14-00157-f002:**
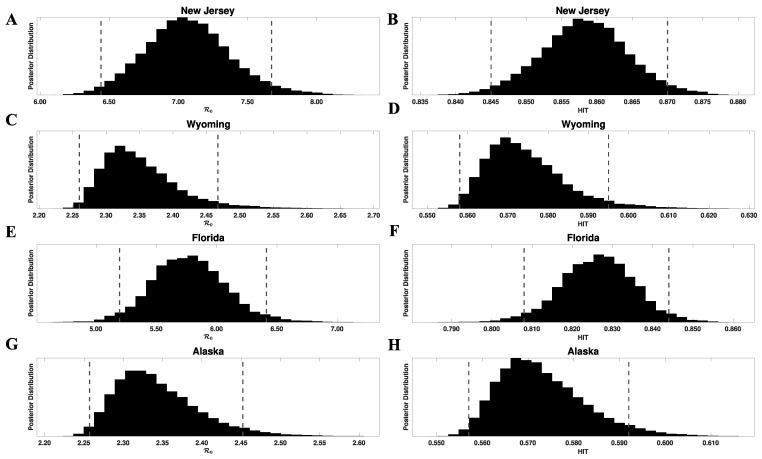
Marginal posterior distributions of ${ℛ}_{0}$ (left panels) and HIT (right panels) for ancestral strains of SARS-CoV-2 in four US states: (**A**,**B**) New Jersey, (**C**,**D**) Wyoming, (**E**,**F**) Florida, and (**G**,**H**) Alaska. Inferences are based on daily reports of new cases from 21 January to 21 June 2020. Each ${ℛ}_{0}$ posterior was obtained from the corresponding marginal posterior for β and Equation (1). Each HIT posterior was obtained from the relation $\mathrm{HIT}=1-1/{ℛ}_{0}$ and the corresponding marginal posterior for ${ℛ}_{0}$. The 95% credible intervals for ${ℛ}_{0}$ are as follows: (6.44, 7.67) for New Jersey, (2.26, 2.47) for Wyoming, (5.20, 6.41) for Florida, and (2.26, 2.45) for Alaska. The 95% credible intervals for the HIT estimates are as follows: (0.84, 0.87) for New Jersey, (0.56, 0.59) for Wyoming, (0.81, 0.84) for Florida, and (0.56, 0.59) for Alaska. For each panel, the endpoints of the corresponding credible interval are indicated with vertical broken lines.

**Figure 3 viruses-14-00157-f003:**
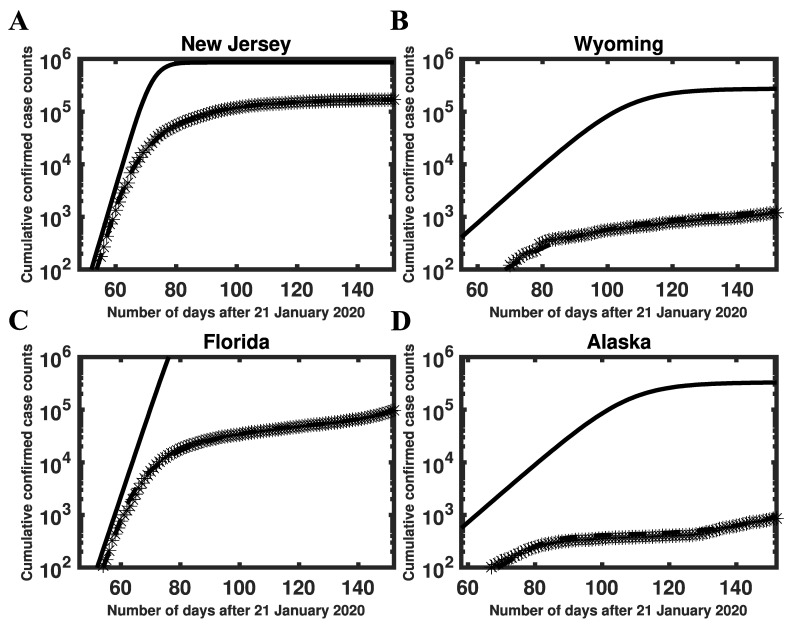
Consistency of model-derived λ estimates with empirical growth rates during initial exponential increase in disease incidence in (**A**) New Jersey, (**B**) Wyoming, (**C**) Florida, and (**D**) Alaska. In each panel, the initial slope of the solid curve corresponds to λ (calculated as described in Materials and Methods), the crosses indicate empirical cumulative case counts, and the broken line is the model prediction based on MAP estimates for adjustable parameters. The solid curve is derived from the reduced model (Equations (1)–(8) in the Appendix A). This curve shows cumulative case counts had there not been any interventions to limit disease transmission. As can be seen, the initial slopes of the solid and broken curves are comparable. We selected n=0 for New Jersey and Wyoming and n=1 for Florida and Alaska. Among 35 states with n=0, New Jersey had the largest inferred λ value (0.45) and Wyoming had the smallest inferred λ value (0.13). Among 15 states with n=1, Florida had the largest inferred value of λ (0.39) and Alaska had the smallest inferred value of λ (0.13). It should be noted that, in contrast with Figure 1, the y-axis here indicates cumulative (vs. daily) number of cases on a logarithmic (vs. linear) scale.

**Figure 4 viruses-14-00157-f004:**
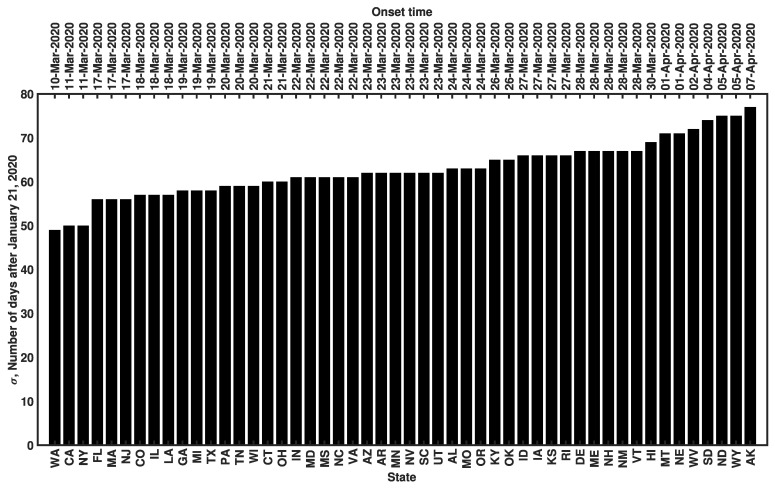
Onset times of COVID-19 disease transmission for ancestral strains of SARS-CoV-2 in all 50 US states. The onset time σ is defined as the first day on which the cumulative reported case count was 200 cases or more. Dates corresponding to σ values on the vertical axis are indicated above each bar. States are indicated using two-letter US postal service state abbreviations (https://about.usps.com/who-we-are/postal-history/state-abbreviations.pdf (accessed on 19 September 2021)).

**Figure 5 viruses-14-00157-f005:**
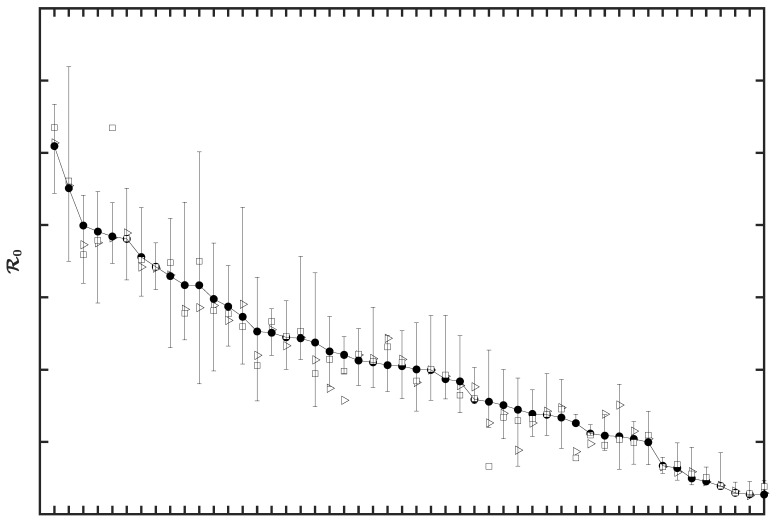
MAP estimates of the basic reproduction number ${ℛ}_{0}$ for ancestral strains of SARS-CoV-2 in all 50 US states. The different symbols refer to different training datasets used to estimate ${ℛ}_{0}$. Open triangles correspond to surveillance data collected from 21 January to 21 May 2020, filled circles correspond to surveillance data collected from 21 January to 21 June 2020, and open squares correspond to surveillance data collected from 21 January to 21 July 2020. Estimates of ${ℛ}_{0}$ are sorted by state from largest to smallest values according to the ${ℛ}_{0}$ estimates derived from the surveillance data collected for 21 January to 21 June 2020. The whiskers associated with each filled circle indicate the 95% credible interval (inferred from the 5-month dataset). States are indicated using two-letter US postal service state abbreviations (https://about.usps.com/who-we-are/postal-history/state-abbreviations.pdf (accessed on 19 September 2021)).

**Figure 6 viruses-14-00157-f006:**
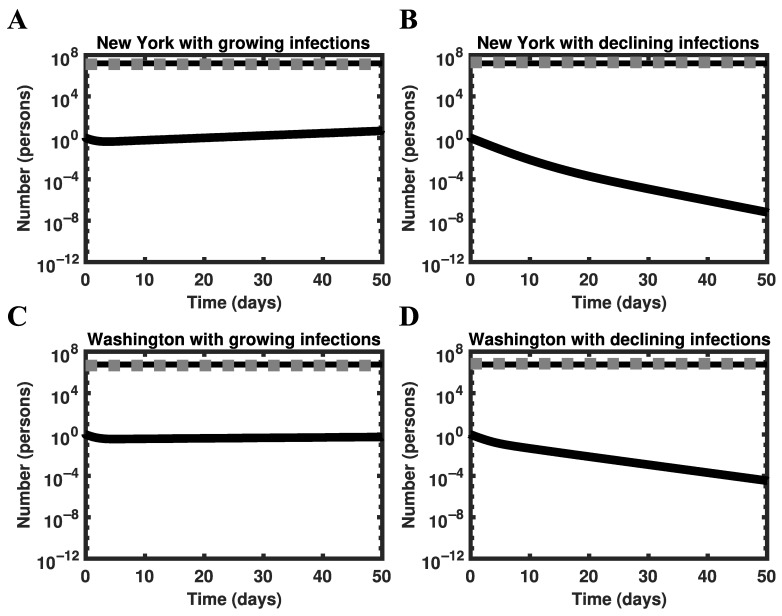
Perturbation analysis using the full model of Lin et al. [22] for the states of New York (panels (**A**,**B**)) and Washington (panels (**C**,**D**)). In each panel, the black solid line represents the number of infectious persons (initially 1), the black broken line represents the threshold number of persons required for herd immunity (i.e., $S_{h}$), and the gray broken line represents the number of recovered persons (initially $S_{h}-\varepsilon$, obtained as described in Results). Simulations are based on MAP estimates for model parameters obtained using surveillance data collected from 21 January to 21 June 2020.

**Figure 7 viruses-14-00157-f007:**
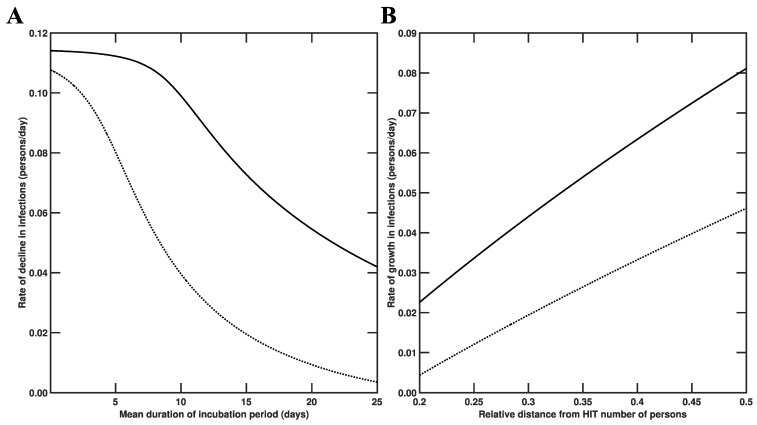
Dependence of disease burden on key model parameters for the states of New York and Washington. In each panel, the solid line corresponds to New York and the broken line corresponds to Washington. In Panel (**A**), rate of decline in infections is plotted as a function of the mean duration of the incubation period (in days), which is obtained as $m/k_{L}$, where m is the number of stages in the incubation period and $k_{L}$ characterizes disease progression from one stage to the next. We take m=5 as in the study of Lin et al. [22]. In Panel (**B**), the rate of growth in infections is plotted as a function of the relative distance from the herd immunity threshold number of persons, which is defined as $\varepsilon /S_{h}$.

**Figure 8 viruses-14-00157-f008:**
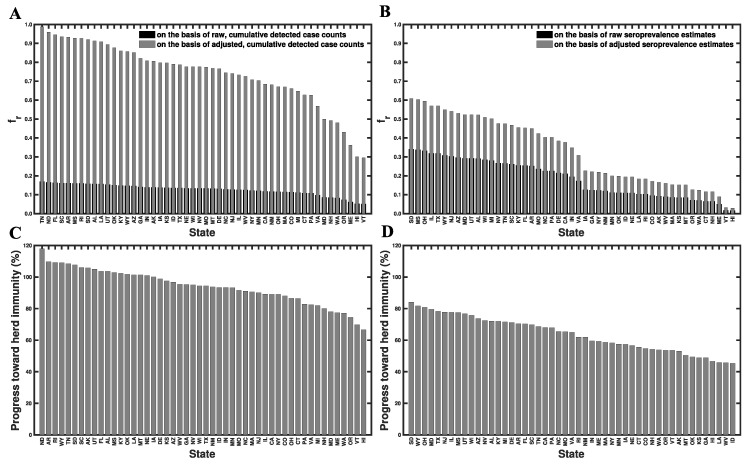
Percent progress toward herd immunity in each of the 50 US states. Percent progress 𝒫 indicates the fraction of immune persons required for herd immunity. 𝒫 was calculated using Equation (2). Black bars (Panel (**A**)) correspond to the first scenario (i.e., $f_{r}$ estimated as the number of detected cases divided by population size), gray bars (Panels (**A**,**C**)) correspond to the second scenario (i.e., $f_{r}$ estimated as the number of detected cases within a population divided by the population size, adjusted for lack of detection of undiagnosed SARS-CoV-2 infections), black bars (Panel (**B**)) correspond to the third scenario (i.e., $f_{r}$ given by seroprevalence survey results), and gray bars (Panels (**B**,**D**)) correspond to the fourth scenario (i.e., $f_{r}$ given by seroprevalence survey results adjusted for lack of detection of asymptomatic cases). Estimates for 𝒫 are sorted by state from largest to smallest values according to the second scenario (Panels (**A**,**C**)) and the fourth scenario (Panels (**B**,**D**)). North Dakota was omitted from Panels (**B**,**D**) because a recent estimate of seroprevalence was not available at Ref. [25]. States are indicated using two-letter US postal service state abbreviations (https://about.usps.com/who-we-are/postal-history/state-abbreviations.pdf (accessed on 19 September 2021)).

**Figure 9 viruses-14-00157-f009:**
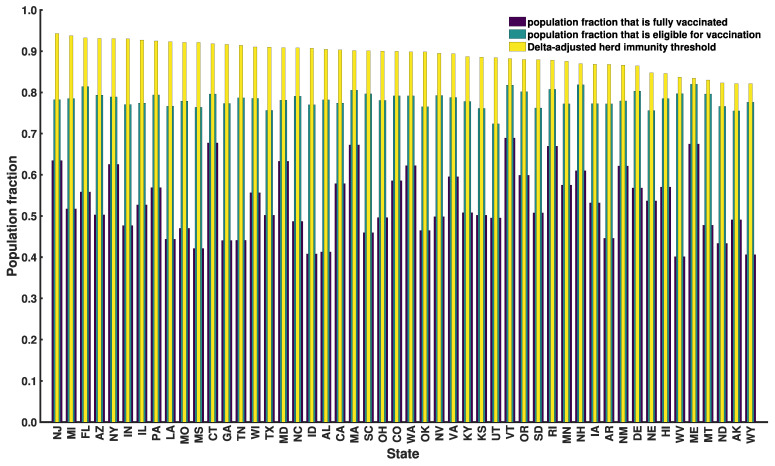
Vaccine eligibility and vaccine coverage in each of the 50 US states on 20 September 2021. Purple bars correspond to vaccine coverage, i.e., the population fraction that is fully vaccinated [24]. Teal bars correspond to vaccine eligibility, i.e., the population fraction that is eligible for vaccination. We estimated the eligible population fraction as the adult fraction of the population [47], i.e., the population fraction 18 years or older. Yellow bars correspond to Delta-adjusted HIT estimates from Appendix A. States are indicated using two-letter US postal service state abbreviations (https://about.usps.com/who-we-are/postal-history/state-abbreviations.pdf (accessed on 19 September 2021)).

**Table 1 viruses-14-00157-t001:** Maximum a posteriori (MAP) estimates and 95% credible intervals for epidemic parameters (β, λ, ${ℛ}_{0}$, HIT, and Delta-adjusted HIT) for the states of New Jersey, Wyoming, Florida, and Alaska.

State	$$\beta \;(d^{-1})$$	$$\lambda \;(d^{-1})\;*$$	$${ℛ}_{0}\;**$$	HIT ***	Delta-Adjusted HIT ****
New Jersey	0.65 (0.59–0.71)	0.45 (0.41–0.48)	7.1 (6.4–7.7)	0.86 (0.84–0.87)	0.94 (0.94–0.95)
Wyoming	0.21 (0.21–0.23)	0.13 (0.13–0.15)	2.3 (2.3–2.5)	0.56 (0.56–0.59)	0.82 (0.82–0.84)
Florida	0.55 (0.48–0.59)	0.39 (0.34–0.41)	6.0 (5.2–6.4)	0.83 (0.81–0.84)	0.93 (0.92–0.94)
Alaska	0.21 (0.21–0.23)	0.13 (0.13–0.14)	2.3 (2.3–2.5)	0.56 (0.56–0.59)	0.82 (0.82–0.84)

In this analysis, we used surveillance data (daily reports of new cases) available from 21 January 2020 to 21 June 2020 (inclusive dates) to estimate parameter values through Bayesian inference. * Computed as described in Appendix A. ** Calculated using Equation (1). *** Obtained through the relation HIT = $1-1/{ℛ}_{0}$. **** Based on Delta being 2.46 times more infectious than ancestral strains.

## Data Availability

Inferences were performed using problem-specific code. The functionality of the code has been added to a freely available open-source software package (PyBioNetFit, version 1.1.9). We have confirmed that the results of the problem-specific code are reproduced by PyBioNetFit. PyBioNetFit is available online (https://github.com/lanl/pybnf (accessed on 29 November 2021).) along with inference job setup files (https://github.com/lanl/PyBNF/tree/master/examples/Mallela2021States (accessed on 29 November 2021).). The EXP files contain the data used in inference.

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
