# Peer review of "Bayesian Inference of State-Level COVID-19 Basic Reproduction Numbers across the United States"

_viruses, 2022, doi:10.3390/v14010157_

Round 1
Reviewer 1 Report
Mallela and colleagues proposed a research article aimed at developing a method to estimate the COVID-19 disease transmission through state-level ?0 estimates obtained using Bayesian inference. Overall, the authors’ model is very rigorous and convincing. Below are reported some minor comments that will improve the quality of the manuscript:
1) Are the following data related to the US population? “As of September 20, 2021, 181,728,072 persons have been fully vaccinated, an additional 30,307,256 persons have been partially vaccinated, and an uncertain number of persons have acquired immunity through infection [2].”. Please clarify. In addition, indicate the percentage of vaccinated population;
2) Please consider and the impact of surveillance diagnostic strategies in the population as a means to early detect and isolate infected people thus artificially reducing the R0 value. For this purpose, briefly describe the current and innovative methods for the diagnosis and the containment of infected individuals, please see:
- PMID: 33846767
- PMID: 32348588
- PMID: 33899035
3) In the Discussion section, the authors have to better explain the differences of R0 and HIT values obtained in different States
Reviewer 2 Report
1.R0 values are different at different states. New Jersey state was the highest. What is the problem? Is it due to different viral strains? Please make discussion.
- The authors used data from January to June 2021. Why not use the data for 2020? The readers will be interesting for the whole pictures of COVID-19 infections in US. If possible, please analyze and add at the revised one.
3.R0 can vary with different strains of viruses. Did the authors analyze based on the time dominant at different viral strain dominant in US and add at the revised manuscript?
- Please add justification for usage of Lin et al model in this study at the revised manuscript. What is the advantage of using his model?
5.The author described that none of the states did not reach herd immunity in US. How about the vaccination coverage? Is it not due to low vaccine efficacy? This is very important for booster dose. Please make discussion and highlight it.
6.In figure5, according to perturbation analysis, the growing infections rate and declining infections rate was different. Is it due to mitigation measures or viral strains? How about other states?
7.At page 10, line 327, there is cover the word “toward” by “B” from Figure-6. Please correct it.
8.. In US, ho about the percentage of children? It can be affect on herd immunity level as children cannot vaccinate with current vaccine? Please make discussion.
